# An Algorithmic Framework For Differentially Private Data Analysis on Trusted Processors

**Joshua Allen**　　　　　**Bolin Ding**$^*$　　　　　**Janardhan Kulkarni**
**Harsha Nori**　　　　　**Olga Ohrimenko**　　　　　**Sergey Yekhanin**

Microsoft

## Abstract

Differential privacy has emerged as the main definition for private data analysis and machine learning. The *global* model of differential privacy, which assumes that users trust the data collector, provides strong privacy guarantees and introduces small errors in the output. In contrast, applications of differential privacy in commercial systems by Apple, Google, and Microsoft, use the *local model*. Here, users do not trust the data collector, and hence randomize their data before sending it to the data collector. Unfortunately, local model is too strong for several important applications and hence is limited in its applicability. In this work, we propose a framework based on trusted processors and a new definition of differential privacy called *Oblivious Differential Privacy*, which combines the best of both local and global models. The algorithms we design in this framework show interesting interplay of ideas from the streaming algorithms, oblivious algorithms, and differential privacy.

## 1　Introduction

Most large IT companies rely on access to raw data from their users to train machine learning models. However, it is well known that models trained on a dataset can release private information about the users that participate in the dataset [13, 50]. With new GDPR regulations and also ever increasing awareness about privacy issues in the general public, doing private and secure machine learning has become a major challenge to IT companies. To make matters worse, while it is easy to spot a violation of privacy when it occurs, it is much more tricky to give a rigorous definition of it.

*Differential privacy (DP)*, introduced in the seminal work of Dwork *et al.* [20], is arguably the only mathematically rigorous definition of privacy in the context of machine learning and big data analysis. Over the past decade, DP has established itself as the defacto standard of privacy with a vast body of research and growing acceptance in industry. Among its many strengths, the promise of DP is intuitive to explain: No matter what the adversary knows about the data, the privacy of a single user is protected from output of the data-analysis. A differentially private algorithm guarantees that the output does not change significantly, as quantified by a parameter $\epsilon$, if the data of any single user is omitted from the computation, which is formalized as follows.

**Definition 1.1.** *A randomized algorithm $\mathcal{A}$ is $(\epsilon, \delta)$-differentially private if for any two neighboring databases $\mathcal{D}_1, \mathcal{D}_2$ any subset of possible outputs $S \subseteq \mathcal{Z}$, we have:*

$$\Pr\left[\mathcal{A}(D_1) \in S\right] \leq e^\epsilon \cdot \Pr\left[\mathcal{A}(D_2) \in S\right] + \delta.$$

---

$^*$Current affiliation: Alibaba Group. Work done while at Microsoft.

This above definition of DP is often called *global differential privacy* (GDP). It assumes that users are willing to trust the data collector. There is a large body of work on GDP, and many non-trivial machine learning problems can be solved in this model very efficiently. See authoritative book by Dwork and Roth [22] for more details. However in the context of IT companies, adoption of GDP is not possible as there is no trusted data collector – users want privacy of their data from the data collector. Because of this, all industrial deployments of DP by Apple, Google, and Microsoft, with the exception of Uber [32], have been set in the so called *local model of differential privacy (LDP)* [24, 19, 18]. In the LDP model, users randomize their data *before* sending it to the data collector.

**Definition 1.2.** *A randomized algorithm $\mathcal{A} : \mathcal{V} \to \mathcal{Z}$ is $\epsilon$-locally differentially private ($\epsilon$-LDP) if for any pair of values $v, v' \in \mathcal{V}$ held by a user and any subset of output $S \subseteq \mathcal{Z}$, we have:*

$$\Pr\left[\mathcal{A}(v) \in S\right] \le e^{\epsilon} \cdot \Pr\left[\mathcal{A}(v') \in S\right].$$

Despite its very strong privacy guarantees, the local model has several drawbacks compared to the global model: many important problems cannot be solved in the LDP setting within a desired level of accuracy. Consider the simple task of understanding the number of distinct websites visited by users, or words in text data. This problem admits no good algorithms in LDP setting, whereas in the global model the problem becomes trivial. Even for problems that can be solved in LDP setting [24, 7, 6, 19], errors and $\epsilon$ are significantly larger compared to the global model. For example, if one is interested in understanding the histogram of websites visited by users, in the LDP setting an optimal algorithm achieves an error of $\Omega(\sqrt{n})$, whereas in the global model error is $O(\frac{1}{\epsilon})$. See experiments and details in [11] for scenarios where the errors introduced by (optimal) LDP algorithms are unacceptable in practice. Finally, in GDP there are several results that give much stronger guarantees than the standard composition theorems: for example, one can answer exponentially many linear queries (even online) using private multiplicative weight update algorithm [21]. Such results substantially increase the practical relevance of GDP algorithms. However, the local model of differential privacy admits no such elegant solutions.

These drawbacks of LPD naturally lead to the following question:

*Are there ways to bridge the local and global differential privacy models such that users enjoy the privacy guarantees of the local model whereas the data collector enjoys the accuracy of the global model?*

This question has attracted a lot of interest in the research community recently. In remarkable recent results, the authors of [5, 15, 23] propose a *secure shuffle* as a way to bridge local and global models of DP. They show that if one has access to a *user anonymization primitive*, and if every user uses a local DP mechanism, then the overall privacy-accuracy trade-off is similar to the global model. However, access to anonymization primitives that users can trust is a difficult assumption to implement in practice, and only shifts the trust boundary. For example, implementing the anonymization primitive via mixnets requires assumption on non-collusion between the mixing servers. Recall that the main reason most companies adopted LDP setting is because users do not trust the data collector.

In this paper, we propose a different approach based on trusted processors (for example, Intel SGX [31]) and a new definition called Oblivious Differential Privacy (ODP) that help to design algorithms that enjoy the privacy guarantees of both local and global models; see Figure 1 (left) for an illustration. Our framework gives the following guarantees.

1. Data is collected, stored, and used in an encrypted form and is protected from the data collector.

2. The data collector obtains information about the data only through the results of a DP-algorithm.

The DP-algorithms themselves run within a Trusted Execution Environments (TEE) that guarantee that the data is decrypted only by the processor during the computation and is always encrypted in memory. Hence, raw data is inaccessible to anyone, including the data collector. To this end, our framework is similar to other systems for private data analysis based on trusted processors, including machine learning algorithms [41] and data analytical platforms such as PROCHLO [11, 48]. Recently, systems for supporting TEEs have been announced by Microsoft[2] and Google [3], and we anticipate a wide adoption of this model for doing private data analysis and machine learning.

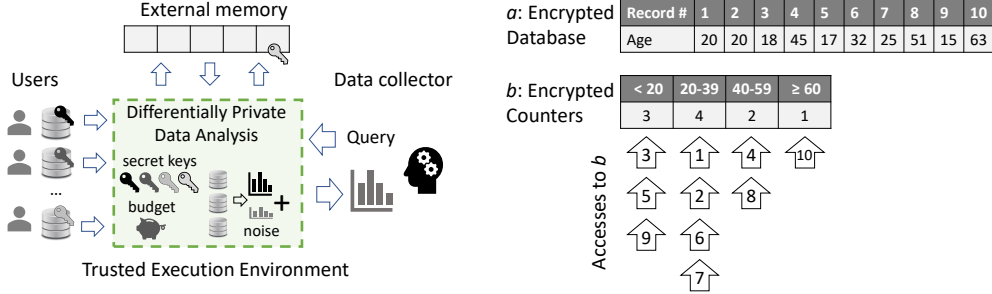

Figure 1: *Left:* Secure differentially-private data-analysis. *Right:* Visualization of the access pattern of a naive histogram computation over a database and four age ranges counters ($k = 4$) stored in arrays $a$ and $b$, respectively. The code reads a record from $a$, decrypts it, accesses the corresponding age bucket in $b$, decrypts its counter, increments it, encrypts it and writes it back. The arrows indicate increment accesses to the histogram counters ($b$) and the numbers correspond to records of $a$ that were accessed prior to these accesses. An adversary of the accesses to $a$ and $b$ learns the histogram and which database records belong to the same age range.

The private data analysis within trusted processors has to be done carefully since the rest of the computational environment is untrusted and is assumed to be under adversary's control. Though the data is encrypted, the adversary can learn private information based on memory access patterns through caches and page faults. In particular, *memory access patterns* are often dependent on private information and have been shown to be sufficient to leak information [42, 10, 55]. Since differentially private algorithms in the global model have been designed with a trusted data collector in mind, they are also susceptible to information leakage through their access patterns.

The main goal of our paper is to formalize the design of differentially private algorithms in the trusted processor environments. Our contributions are summarized below:

- Building on the recent works of [41, 11], we propose a framework that enables collection and analysis of data in the global model of differential privacy without relying on a trusted curator. Our framework uses encryption and secure processors to protect data and computation such that only the final differentially private output of the computation is revealed.
- Trusted execution environments impose certain restrictions on the design of algorithms. We formalize the mathematical model for designing differentially private algorithms in TEEs.
- We define a new class of differentially-private algorithms (Definition 3.1) called *obliviously differentially private algorithms* (ODP), which ensure that privacy leakage that occurs through algorithm's memory access patterns and the output together satisfy the DP guarantees.
- We design ODP algorithms with provable performance guarantees for some commonly used statistical routines such as computing the number of distinct elements, histogram, and heavy hitters. We prove that the privacy and error guarantees of our algorithms (Theorems (4.1, 4.4,4.5) are significantly better than in the local model, and obliviousness does not come at a steep price.

A technically novel aspect of our paper is that it draws ideas from various different fields: streaming algorithms, oblivious algorithms, and differentially private algorithms. This fact becomes clear in §4 where we design ODP algorithms.

**Related work**    There are several systems that propose confidential data analysis using TEEs [41, 56, 48, 11]. PROCHLO [11], in particular, provides support for differential private data analysis. While PROCHLO emphasizes more on the system aspects (without formal proofs), our work gives a formal framework based on oblivious differential privacy for analyzing and designing algorithms for private data analysis in TEEs. Oblivious sampling algorithms proposed in [47] generate samples securely s.t. privacy amplification can be used when analyzing DP algorithms executed on the samples in TEE.

## 2    Preliminaries

### 2.1    Secure Data Analysis with Trusted Processors and Memory Access Leakage

A visualization of our framework is given in Figure 1 (left). We use Intel Software Guard Extensions (SGX) as an example of a trusted processor. Intel SGX [31] is a set of CPU instructions that allows

user-level code to allocate a private region of memory, called an enclave (which we also refer to as a TEE), which is accessible only to the code running in an enclave. The enclave memory is available in raw form only inside the physical processor package, but it is encrypted and integrity protected when written to memory. As a result, the code running inside of an enclave is isolated from the rest of the system, including the operating system. Additionally, Intel SGX supports software attestation [3] that allows the enclave code to get messages signed with a private key of the processor along with a digest of the enclave. This capability allows users to verify that they are communicating with a specific piece of software (i.e., a differentially-private algorithm) running in an enclave hosted by the trusted hardware. Once this verification succeeds, the user can establish a secure communication channel with the enclave (e.g., using TLS) and upload data. When the computation is over, the enclave, including the local variables and data, is deleted.

An enclave can access data that is managed by the trusted processor (e.g., data in registers and caches) or by the software that is not trusted (e.g., an operating system). As a result, in the latter case, data in the external memory has to be encrypted and integrity protected by the code running inside of an enclave. Unfortunately, encryption and integrity are not sufficient to protect against the adversary described in the introduction that can see the addresses of the data being accessed even if the data is encrypted. There are several ways the adversary can extract the addresses, i.e., the *memory access pattern*. Some typical examples are: an adversary with physical access to a machine can attach probes to a memory bus, an adversary that shares the same hardware as the victim enclave code (e.g., a co-tenant) can use shared resources such as caches to observe cache-line accesses, while a compromised operating system can inject page faults and observe page-level accesses. Memory access patterns have been shown to be sufficient to extract secrets and data from cryptographic code [34, 43, 10, 45, 42], from genome indexing algorithms [12], and from image and text applications [55]. (See Figure 1 (right) for a simple example of what can be extracted by observing accesses of a histogram computation.) As a result, accesses leaked through memory side-channels [4] undermine the confidentiality promise of enclaves [55, 49, 30, 35, 17, 12, 38].

## 2.2 Data-Oblivious Algorithms

Data-oblivious algorithms [26, 41, 52, 27] are designed to protect memory addresses against the adversary described in §2.1: they produce data access patterns that appear to be independent of the sensitive data they compute on. They can be seen as external-memory algorithms that perform computation inside of small private memory while storing the encrypted data in the external memory and accessing it in a *data-independent* manner. We formally capture this property below. Suppose external memory is represented by an array $a[1, 2, ..., M]$ for some large value of $M$.

**Definition 2.1** (Access pattern). *Let* $\mathsf{op}_j$ *be either a* $\mathsf{read}(\mathsf{a}[i])$ *operation that reads data from the location* $a[i]$ *to private memory or a* $\mathsf{write}(\mathsf{a}[i])$ *operation that copies some data from the private memory to the external memory* $a[i]$. *Then, let* $s := (\mathsf{op}_1, \mathsf{op}_2, \ldots, \mathsf{op}_t)$ *denote an access pattern of length* $t$ *of algorithm* $\mathcal{A}$ *to the external memory.*

Note that the adversary can see only the addresses accessed by the algorithm and whether it is a read or a write. It cannot see the data since it is encrypted using probabilistic encryption that guarantees that the adversary cannot tell if two ciphertexts correspond to the same record or two different ones.

**Definition 2.2** (Data-oblivious algorithm). *An algorithm* $\mathcal{A}$ *is data-oblivious if for any two inputs* $I_1$ *and* $I_2$, *and any subset of possible memory access patterns* $S \subseteq \mathcal{S}$, *where* $\mathcal{S}$ *is the set of all possible memory access patterns produced by an algorithm, we have:*

$$\Pr\left[\mathcal{A}(I_1) \in S\right] = \Pr\left[\mathcal{A}(I_2) \in S\right]$$

It is instructive to compare this definition with the definition of differential privacy. The definition of oblivious algorithms can be thought of as a generalization of DP to memory access patterns, where $\epsilon = 0$ and the guarantee should hold even for *non-neighboring* databases. Similar to external memory algorithms, the overhead of a data-oblivious algorithm is measured in the number of accesses it makes to external memory, while computations on private memory are assumed to be constant. Some algorithms naturally satisfy Definition 2.2 while others require changes to how they operate. For example, scanning an array is a data-oblivious algorithm since for any array of the same size every element of the array is accessed. Sorting networks [9] are also data-oblivious as element identifiers

accessed by compare-and-swap operations are fixed based on the size of the array and not its content. On the other hand, quicksort is not oblivious as accesses depend on the comparison of the elements with the pivot element. As can be seen from Figure 1 (right), a naive histogram algorithm is also not data-oblivious. (See the supplementary material for overheads of several oblivious algorithms.)

In this paper, we focus on measuring the overhead in performance in terms of the number of memory accesses of oblivious algorithms. We omit the cost of setting up a TEE, which is a one-time cost proportional to the size of the code and data loaded in a TEE, and the cost of encryption and decryption, which is linear in the size of the data and is often implemented in hardware.

**Oblivious RAM (ORAM)** is designed to hide the indices of accesses to an array of size $n$, i.e., it hides how many times and when an index was accessed. There is a naive and inefficient way to hide an access by reading and writing to every index. Existing ORAM constructions incur sublinear overhead by using specialized data structures and re-arranging the external memory [26, 29, 46, 51, 54, 28]. The best known ORAM construction has $O(\log n)$ [4] overhead. Since it incurs high constants, Path ORAM [52] with the overhead of $O((\log n)^2)$ is a preferred option in practice. ORAM can be used to transform any RAM program whose number of accesses does not depend on sensitive content; otherwise, the number of accesses needs to be padded. However, if one is willing to reveal the algorithm being performed on the data, then for some computations the overhead of specialized constructions can be asymptotically lower than of the one based on ORAM.

**Data-oblivious shuffle [40]** takes as a parameter an array $a$ (stored in external memory) of size $n$ and a permutation $\pi$ and permutes $a$ according to $\pi$ such that $\pi$ is not revealed to the adversary. The Melbourne shuffle [40] is a randomized data-oblivious shuffle that makes $O(cn)$ deterministic accesses to external memory, assuming private memory of size $\sqrt[c]{n}$, and fails with negligible probability. For example, for $c = 2$ the overhead of the algorithm is constant as any non-oblivious shuffle algorithm has to make at least $n$ accesses. The Melbourne shuffle with smaller private memories of size $m = \omega(\log n)$ incurs slightly higher overhead of $O(n \log n / \log m)$ as showed in [44]. We will use oblivious_shuffle($a$) to refer to a shuffle of $a$ according to some random permutation that is hidden from the adversary.

Note that the user anonymization primitive that the shuffle model of differential privacy [5, 15, 23] relies on can be implemented in TEEs with a data-oblivious shuffle [11]. However, in this case the trust model of the shuffle model DP will be the same as described in the next section.

# 3 Algorithmic Framework and Oblivious Differential Privacy

We now introduce the definition of Oblivious Differential Privacy (ODP), and give an algorithmic framework for the design of ODP-algorithms for a system based on trusted processors (§2.1). As we mentioned earlier, off-the-shelf DP-algorithms may not be suitable for TEEs for two main reasons.

**Small Private Memory:** The private memory, which is protected from the adversary, available for an algorithm within a trusted processor is much smaller than the data the algorithm has to process. A reasonable assumption on the size of private memory is *polylogarithmic* in the input size.

**Access Patterns Leak Privacy:** An adversary who sees memory access patterns of an algorithm to external memory can learn useful information about the data, compromising the differential privacy guarantees of the algorithm.

Therefore, the algorithm designer needs to guarantee that memory access patterns do not reveal any private information, and the overall algorithm is differentially private [5]. *To summarize, in our attacker model private information is leaked either by the output of a DP algorithm or through memory access patterns.* We formalize this by introducing the notion of *Oblivious Differential Privacy*, which combines the notions of differential privacy and oblivious algorithms.

**Definition 3.1.** *Let $D_1$ and $D_2$ be any two neighboring databases that have exactly the same size $n$ but differ in one record. A randomized algorithm $\mathcal{A}$ that has small private memory (i.e., sublinear in $n$) and accesses external memory is $(\epsilon, \delta)$-obliviously differentially private (ODP), if for any subset of possible memory access patterns $S \subseteq \mathcal{S}$ and any subset of possible outputs $O$ we have:*

$$\Pr\left[\mathcal{A}(D_1) \in (O, S)\right] \leq e^{\epsilon} \cdot \Pr\left[\mathcal{A}(D_2) \in (O, S)\right] + \delta.$$

We believe that the above definition gives a systematic way to design DP algorithms in TEE settings. An algorithm that satisfies the above definition guarantees that the private information released through output of the algorithm *and* through the access patterns is quantified by the parameters $(\epsilon, \delta)$. Similar to our definition, Wagh *et al.* [53] and more recently, in a parallel work, Chan *et al.* [14] also consider relaxing the definition of obliviousness for hiding access patterns from an adversary. However, their definitions serve complementary purpose to ours: *they apply DP to oblivious algorithms, whereas we apply obliviousness to DP algorithms.* This is crucial since algorithms that satisfy the definition in [53, 14] may not satisfy DP when the output is released, which is the main motivation for using differentially private algorithms. Our results together with [14] highlight that DP and oblivious algorithms is an interesting area for further research for private and secure ML.

*Remarks:* In the real world, the implementation of a TEE relies on cryptographic algorithms (e.g., encryption and digital signatures) that are computationally secure and depend on a security parameter of the system. As a result any differentially private algorithm operating inside of a TEE has a non-zero parameter $\delta$ that is negligible in the security parameter.

In the paper, we only focus on memory accesses; but our definitions and framework can be easily extended to other forms of side-channel attacks such as timing attacks (e.g., by incorporating the time of each access), albeit requiring changes to algorithms presented in the next section to satisfy them.

**Connections to Streaming Algorithms:** One simple strategy to satisfy Definition 3.1 is to take a DP-algorithm and guarantee that every time the algorithm makes an access to the public memory, it makes a pass over the entire data. However, such an algorithm incurs a multiplicative overhead of $n$ on the running time, and the goal would be to minimize the number of passes made over the data. Interestingly, these algorithms precisely correspond to the *streaming algorithms*, which are widely studied in big-data analysis. In the streaming setting, one assumes that we have only $O(\log n)$ bits of memory and data stream consists of $n$ items, and the goal is to compute functions over the data. Quite remarkably, several functions can be approximated very well in this model. See [39] for an extensive survey. Since there is a large body of work on streaming algorithms, we believe that many algorithmic ideas there can be used in the design of ODP algorithms. We give example of such algorithms for distinct elements §4.1 and heavy hitters problem in §4.3.

**Privacy Budget:** Finally, we note that the system based on TEEs can support interactive data analysis where the privacy budget is hard-coded (and hence verified by each user before they supply their private data). The data collector's queries decrease the budget appropriately and the code exits when privacy budget is exceeded. Since the budget is maintained as a local variable within the TEE it is protected from replay attacks while the code is running. If TEE exits, the adversary cannot restart it without notifying the users since the code requires their secret keys. These keys cannot be used for different instantiations of the same code as they are also protected by TEE and are destroyed on exit.

# 4 Obliviously Differentially Private Algorithms

In this section, we show how to design ODP algorithms for the three most commonly used statistical queries: counting the number of distinct elements in a dataset, histogram of the elements, and reporting heavy hitters. The algorithms for these problems exhibit two common themes: 1) For many applications it is possible to design DP algorithms without paying too much overhead to enforce obliviousness. 2) The interplay of ideas from the streaming and oblivious algorithms literature in the design of ODP algorithms.

Before we continue with the construction of ODP algorithms, we make a subtle but important point. Recall that in our Definition 3.1, we require that two neighboring databases have exactly the same size. If the neighboring databases are of different sizes, then the access patterns can be of different lengths, and it is impossible to satisfy the ODP-definition. This definition does not change the privacy guarantees as in many applications the size of the database is known in advance; e.g., number of users of a system. However, it has implications on the sensitivity of the queries. For example, histogram queries in our definition have sensitivity of 2 where in the standard definition it is 1.

## 4.1 Number of Distinct Items in a Database

As a warm-up for the design of ODP algorithms, we begin with the distinct elements problem. Formally, suppose we are given a set of $n$ users and each user $i$ holds an item $v_i \in \{1, 2, ..., m\}$,

where $m$ is assumed to be much larger than $n$. This is true if a company wants to understand the number of distinct websites visited by its users or the number of distinct words that occur in text data. Let $n_v$ denote the number of users that hold the item $v$. The goal is to estimate $n^* := |\{v : n_v > 0\}|$.

We are not aware of a reasonable solution that achieves an additive error better than $\Omega(n)$ for this problem in the LDP setting. In a sharp contrast, the problem becomes very simple in our framework. Indeed, a simple solution is to do an *oblivious sorting* [1, 9] of the database elements, and then count the number of distinct elements by making another pass over the database. Finally, one can add Laplace noise with the parameter $\frac{1}{\epsilon}$, which will guarantee that our algorithm satisfies the definition of ODP. This is true as a) the sensitivity of the query is 1, as a single user can increase or decrease the number of distinct elements by at most 1; b) we do oblivious sorting. Furthermore, the expected (additive) error of such an algorithm is $1/\epsilon$. Recall that $n^*$ denotes the number of distinct elements in a database. Thus we get:

**Theorem 4.1.** *There exists an oblivious sorting based $(\epsilon, 0)$-ODP algorithm for the problem of finding the number of distinct elements in a database that runs in time $O(n \log n)$. With probability at least $1 - \theta$, the number of distinct elements output by our algorithm is $(n^* \pm \log(1/\theta)\frac{1}{\epsilon})$.*

While above algorithm is optimal in terms of error, we propose a more elegant *streaming algorithm* that does the entire computation in the private memory. The main idea is to use a *sketching technique* to maintain an approximate count of the distinct elements in the private memory and report this approximate count by adding noise from $\mathrm{Lap}(1/\epsilon)$. This will guarantee that our algorithm is $(\epsilon, 0)$-ODP, as the entire computation is done in the private memory and the Laplace mechanism is $(\epsilon, 0)$-DP. There are many streaming algorithms (e.g., Hyperloglog) [25, 33] which achieve $(1 \pm \alpha)$-approximation factor on the number of distinct elements with a space requirement of $\mathrm{polylog}(n)$. We use the following (optimal) algorithm in [33].

**Theorem 4.2.** *There exists a streaming algorithm that gives a $(1 \pm \alpha)$ multiplicative approximation factor to the problem of finding the distinct elements in a data stream. The space requirement of the algorithm is at most $\frac{\log n}{\alpha^2} + (\log n)^2$ and the guarantee holds with probability $1 - 1/n$.*

It is easy to convert the above algorithm to an ODP-algorithm by adding noise sampled from $\mathrm{Lap}(\frac{1}{\epsilon})$.

**Theorem 4.3.** *There exists a single pass (or online) $(\epsilon, 0)$-ODP algorithm for the problem of finding the distinct elements in a database. The space requirement of the algorithm is at most $\frac{\log n}{\alpha^2} + (\log n)^2$. With probability at least $1 - 1/n - \theta$, the number of distinct elements output by our algorithm is $(1 \pm \alpha)n^* \pm \log(1/\theta)\frac{1}{\epsilon}$.*

The additive error of $\pm \log(1/\theta)\frac{1}{\epsilon}$ is introduced by the Laplace mechanism, and the multiplicative error of $(1 \pm \alpha)$ is introduced by the sketching scheme. Although this algorithm is not optimal in terms of the error compared to Theorem 4.1, it has the advantage that it can maintain the approximate count in an *online* fashion.

## 4.2 Histogram

Let $D$ be a database with $n$ records. We assume that each record in the database has a unique identifier. Let $\mathcal{D}$ denote all possible databases of size $n$. Each record (or element) $r \in \mathcal{D}$ has a *type*, which, without loss of generality, is an integer in the set $\{1, 2, \ldots, k\}$.

For a database $D \in \mathcal{D}$, let $n_i$ denote the number of elements of type $i$. Then the histogram function $h : \mathcal{D} \to \mathbb{R}^k$ is defined as $h(D) := (n_1, n_2, \ldots, n_k)$.

A simple differentially private histogram algorithm $\mathcal{A}_{\mathsf{hist}}$ returns $h(D) + (X_1, X_2, \ldots, X_k)$ where $X_i$ are i.i.d. random variables drawn from $\mathrm{Lap}(2/\epsilon)$. This algorithm is *not* obliviously differentially private as the access pattern reveals to the adversary much more information about the data than the actual output. In this section, we design an ODP algorithm for the histogram problem. Let $\hat{n}_i$ denote the number of elements of type $i$ output by our histogram algorithm. We prove the following theorem in this paper.

**Theorem 4.4.** *For any $\epsilon > 0$, there is an $(\epsilon, \frac{1}{n^2})$-ODP algorithm for the histogram problem that runs in time $O(\tilde{n} \log \tilde{n} / \log \log \tilde{n})$ where $\tilde{n} = \max(n, k \log n/\epsilon)$. With probability $1 - \theta$, it holds that*

$$\max_i |\hat{n}_i - n_i| \leq \log(k/\theta) \cdot \frac{2}{\epsilon}.$$

Observe that our algorithm achieves same error guarantee as that of global DP algorithm without much overhead in terms of running time.

To prove that our algorithm is ODP, we need to show that the distribution on the access patterns produced by our algorithm for any two neighboring databases is approximately the same. The same should hold true for the histogram output by our algorithm. We achieve this as follows. We want to use the simple histogram algorithm that adds Laplace noise with parameter $2/\epsilon$, which we know is $\epsilon$-DP. This is true since the histogram queries have sensitivity of 2; that is, if we change the record associated with the single user, then the histogram output changes for at most 2 types. Note that if the private memory size is larger than $k$, then the task is trivial. One can build the entire DP-histogram in the private memory by making a single pass over the database, which clearly satisfies our definition of ODP. However, in many applications $k \gg O(\log n)$. A common example is a histogram on bigrams of words which is commonly used in text prediction. When $k \gg \log n$ the private memory is not enough to store the entire histogram, and we need to make sure that memory access patterns do not reveal too much information to the adversary.

One can make the naive histogram algorithm satisfy Definition 3.1 by accessing the entire public memory for every read/write operation, incurring an overhead of $O(nk)$. Another method to solve the histogram problem would be to rely on oblivious sorting algorithms. The overhead of this method would be the overhead of sorting. However, sorting is usually an expensive operation in practice. Here, we give an arguably simpler and faster algorithm for larger values of $k$ that satisfies the definition of ODP. (At a high level our algorithm is similar to the one which appeared in the independent work by Mazloom and Gordon [36], who use a differentially private histogram to protect access patterns of graph-parallel computation based on garbled circuits, as a result requiring a different noise distribution and shuffling algorithm.)

We give a high-level overview of the algorithm here, and defer the pseudo-code and analysis to the supplementary material. Let $T = n + 20k \log n/\epsilon$.

1. Sample $k$ random variables $X_1, X_2, \ldots, X_k$ from $\text{Lap}(2/\epsilon)$. If any $|X_i| > 10 \log n/\epsilon$, then we set $X_i = 0$ for all $i = 1, 2, .., k$. For all $i$, set $X_i = \lceil X_i \rceil$.
2. We create $(10 \log n/\epsilon + X_i)$ *fake* records of type $i$ and append it to the database $D$. This step together with step 1 ensures that $(10 \log n/\epsilon + X_i)$ is always positive. The main reason to restrict the Laplace distribution's support to $[-10 \log n/\epsilon, 10 \log n/\epsilon]$ is to ensure that we only have positive noise. If the noise is negative, we cannot create fake records in the database simulating this noise.
3. Next, we create $(10k \log n/\epsilon - \sum_i X_i)$ *dummy* records in the database $D$, which do not correspond to any particular type in $1..k$. The dummy records have type $k + 1$. The purpose of this step is to ensure that the length of the output is exactly $T$.
4. Let $\hat{D}$ be the augmented database that contains both dummy and fake records, where the adversary cannot distinguish between database, dummy and fake records as they are encrypted using probabilistic encryption. Obliviously shuffle $\hat{D}$ [40] so that the mapping of records to array $a[1, 2, ..., T]$ is uniformly distributed.
5. Initialise $b$ with $k$ zero counters in external memory. Scan every element from the array $a[1, 2, ..., T]$ and increment the counter in histogram $b$ associated with type of $a[i]$. If the record corresponds to a dummy element, then access the array $b[1, 2, ..., k]$ in *round-robin fashion* and do a fake write without modifying the actual content of $b$.

In the full version [2], we show that above algorithm is $(\epsilon, \frac{1}{n^2})$-ODP for any $\epsilon > 0$. While the proof is long, intuition is simple: The shuffle operation gives a uniform distribution on how $\hat{D}$ is stored in public memory. Since our algorithm is deterministic after the shuffle operation – it just makes a single pass over $\hat{D}$ – the only information adversary learns from the memory access patterns is the histogram of the elements + noise. Now, Laplace noise guarantees that it is DP.

### 4.3 Heavy Hitters

As a final example, we consider the problem of finding frequent items, also called the heavy hitters problem, while satisfying the ODP definition. In this problem, we are given a database $D$ of $n$ users, where each user holds an item from the set $\{1, 2, ..., m\}$. In typical applications such as finding the most frequent websites or finding the most frequently used words in a text corpus, $m$ is usually much

larger than $n$. Hence reporting the entire histogram on $m$ items is not possible. In such applications, one is interested in the list of $k$ most frequent items, where we define an item as frequent if it occurs more than $n/k$ times in the database. In typical applications, $k$ is assumed to be a constant or sublinear in $n$. The problem of finding the heavy hitters is one of the most widely studied problems in the LDP setting [8, 6]. In this section, we show that the heavy hitters problem becomes very simple in our model. Let $\hat{n}_i$ denote the number of occurrences of the item $i$ output by our algorithm and $n_i$ denotes the true count. We give the full proof of the theorem in the supplementary material.

**Theorem 4.5.** *Let $\tau > 1$ be some constant, and let $\epsilon > 0$ be the privacy parameter. Suppose $n/k > \frac{\tau}{\epsilon} \log m$. Then, there exists a $(\epsilon, \frac{1}{m^{\tau-1}})$-ODP algorithm for the problem of finding the top $k$ most frequent items that runs in time $O(n \log n)$. Furthermore, for every item $i$ output by our algorithm it holds that i) with probability at least $(1 - \theta)$, $|\hat{n}_i - n_i| \leq \log(m/\theta) \cdot \frac{2}{\epsilon}$ and ii) $n_i \geq n/k - \log(m/\theta) \cdot \frac{2}{\epsilon}$.*

We remark that the $k$ items output by an ODP-algorithm do not exactly correspond to the top $k$ items due to the additive error introduced by the algorithm. We can use the above theorem to return a list of *approximate heavy hitters*, which satisfies the following guarantees: 1) Every item with frequency higher than $n/k$ is in the list. 2) No item with frequency less than $n/k - 2\log(m/\theta) \cdot \frac{2}{\epsilon}$ is in the list.

We contrast the bound of this theorem with the bound one can obtain in the LDP setting. An *optimal* LDP algorithm can only achieve a guarantee of $|\hat{n}_i - n_i| \leq \sqrt{n \cdot \log(n/\theta) \cdot \frac{\log m}{\epsilon}}$. We refer the reader to [8, 6] for more details about the heavy hitters problem in the LDP setting. For many applications such as text mining, finding most frequently visited websites within a sub-population, this difference in the error turns out to be significant. See experiments and details in [11].

Our algorithm for Theorem 4.5 proceeds as follows: It sorts the elements in the database by type using oblivious sorting. It then initialises an encrypted list $b$ and fills it in while scanning the sorted database as follows. It reads the first element and saves in private memory its type, say $i$, and creates a counter set to 1. It then appends to $b$ a tuple: type $i$ and the counter value. When reading the second database element, it compares its type, say $i'$, to $i$. If $i = i'$, it increments the counter. If $i \neq i'$, it resets the counter to 1 and overwrites the type saved in private memory to $i'$. In both cases, it then appends to $b$ another tuple: the type and the counter from private memory. It proceeds in this manner for the rest of the database. Once finished, it makes a backward scan over $b$. For every new type it encounters, it adds $\mathrm{Lap}(2/\epsilon)$ to the corresponding counter and, additionally, extends the tuple with a flag set to 0. For all other tuples, a flag set to 1 is added instead. It then sorts $b$: by the flag in ascending order and by differentially-private counter values in descending order.

Let $n^*$ be the number of distinct elements in the database. Then the first $n^*$ tuples of $b$ hold all the distinct types of the database together with their differentially-private frequencies. Since these elements are sorted, one can make a pass, returning the types of the top $k$ most frequent items with the highest count (which includes the Laplace noise). Although this algorithm is not $(\epsilon, 0)$-ODP, it is easy to show that it is $(\epsilon, \frac{1}{m^{\tau-1}})$-ODP when $n/k > \tau \log m$, which is the case in all applications of the heavy hitters. Indeed, in most typical applications $k$ is a constant. The proof that the algorithm satisfies the statement of Theorem 4.5 appears in the supplemental material.

**Frequency Oracle Based on Count-Min Sketch** Another commonly studied problem in the context of heavy hitters is the frequency oracle problem. Here, the goal is to answer the number of occurrences of an item $i$ in the database. While this problem can be solved by computing the answer upon receiving a query and adding Laplace noise, there is a simpler approach which might be sufficient in many applications. One can maintain a count-min sketch, a commonly used algorithm in the streaming literature [16], of the frequencies of items by making a single pass over the data. An interesting aspect of this approach is that entire sketch can be maintained in the private memory, hence one does not need to worry about obliviousness. Further, entire count-min sketch can be released to the data collector by adding Laplace noise. An advantage of this approach is that the data collector can get the frequencies of any item he wants by simply referring to the sketch, instead of consulting the DP algorithm. It would be interesting to find more applications of the streaming techniques in the context of ODP algorithms.

## Footnotes

[2]"Introducing Azure confidential computing", accessed October 26, 2019.

[3]"Introducing Asylo: an open-source framework for confidential computing", accessed October 26, 2019.

[4]This should not be confused with vulnerabilities introduced by floating-point implementations [37].

[5]Here, data collector runs algorithms on premise. See the supplementary material for restrictions in the cloud setting.

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
