[Supplementary Material · camera-ready-full.pdf]

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

One can make the naive histogram algorithm satisfy Definition 3.1 by accessing the entire public memory for every read/write operation, incurring an overhead of $O(nk)$. For $k = \text{polylog}(n)$, one can access the histogram array through an oblivious RAM. (See appendix for more details.) Oblivious sorting algorithms could also be used to solve the histogram problem (see §4.3 and §B). However, sorting is usually an expensive operation in practice. Here, we give an arguably simpler and faster algorithm for larger values of $k$ that satisfies the definition of ODP. (At a high level our algorithm is similar to the one which appeared in the independent work by Mazloom and Gordon [36], who use a differentially private histogram to protect access patterns of graph-parallel computation based on garbled circuits, as a result requiring a different noise distribution and shuffling algorithm.)

We first give a high-level overview of the algorithm. The pseudo-code is given in Algorithm 1. Let $T = n + 20k\log n/\epsilon$.

1. Sample $k$ random variables $X_1, X_2, \ldots, X_k$ from $\text{Lap}(2/\epsilon)$. If any $|X_i| > 10\log n/\epsilon$, then we set $X_i = 0$ for all $i = 1, 2, .., k$. For all $i$, set $X_i = \lceil X_i \rceil$.

2. We create $(10\log n/\epsilon + X_i)$ *fake* records of type $i$ and append it to the database $D$. This step together with step 1 ensures that $(10\log n/\epsilon + X_i)$ is always positive. The main reason to restrict the Laplace distribution's support to $[-10\log n/\epsilon, 10\log n/\epsilon]$ is to ensure that we only have positive noise. If the noise is negative, we cannot create fake records in the database simulating this noise.

3. Next, we create $(10k\log n/\epsilon - \sum_i X_i)$ *dummy* records in the database $D$, which do not correspond to any particular type in $1..k$. The dummy records have type $k + 1$. The purpose of this step is to ensure that the length of the output is exactly $T$.

4. Let $\hat{D}$ be the augmented database that contains both dummy and fake records, where the adversary cannot distinguish between database, dummy and fake records as they are encrypted using probabilistic encryption. Obliviously shuffle $\hat{D}$ [40] so that the mapping of records to array $a[1, 2, ..., T]$ is uniformly distributed.

5. Initialise $b$ with $k$ zero counters in external memory. Scan every element from the array $a[1, 2, ..., T]$ and increment the counter in histogram $b$ associated with type of $a[i]$. If the record corresponds to a dummy element, then access the array $b[1, 2, ..., k]$ in *round-robin fashion* and do a fake write without modifying the actual content of $b$.

We shall show that above algorithm is $(\epsilon, \frac{1}{n^2})$-differentially private for any $\epsilon > 0$. Towards that we need the following simple lemma for our proofs.

**Algorithm 1** Oblivious Differentially Private Histogram $\mathcal{A}_{\text{hist}}^{\text{ODP}}(D, k)^6$

$\hat{D} \leftarrow \text{add\_fake\_dummy}(D, k)$
$D' \leftarrow \text{oblivious\_shuffle}(\hat{D})$
$b \leftarrow \{0\}^k$
$\text{ptr} \leftarrow 1$
**for** $r \in D'$ **do**
   $i \leftarrow \text{get\_type}(r)$
   **if** $i = k + 1$ **then**
      $b_{\text{ptr}} \leftarrow b_{\text{ptr}} + 0$
      $\text{ptr} \leftarrow \text{ptr} \mod k + 1$
   **else**
      $b_i \leftarrow b_i + 1$
   **end if**
**end for**
**for** $i \in 1 \dots k$ **do**
   $\hat{b}_i \leftarrow b_i - 10 \log n / \epsilon$
**end for**
**return** $\hat{b}$

**Algorithm 2** Utility procedure for Algorithm 1

**procedure** add\_fake\_dummy$(D, k)$
   **for** $i \in 1 \dots k$ **do**
      $X_i \leftarrow$ *draw a sample from* $\text{Lap}(\frac{2}{\epsilon})$
   **end for**
   *If* $\exists i \; |X_i| > 10 \log n / \epsilon$: $\forall i X_i \leftarrow 0$
   $\forall i, X_i \leftarrow \lceil X_i \rceil$
   **for** $i \in 1 \dots k$ **do**
      **for** $j \in 1 \dots (10 \log n / \epsilon + X_i)$ **do**
         $r' \leftarrow \text{set\_type}(\text{dummy}, i)$
         $D.\text{append}(r')$
      **end for**
   **end for**
   $L \leftarrow 10k \cdot \log n / \epsilon - \sum_{i=1}^{k} X_i$
   **for** $i \in 1 \dots L$ **do**
      $r' \leftarrow \text{set\_type}(\text{dummy}, k + 1)$
      $D.\text{append}(r')$
   **end for**
   **return** $D$

**Lemma 4.1.** $\Pr\left[\max_{1 \le i \le k} |X_i| \ge \frac{10 \log n}{\epsilon}\right] \le \frac{1}{n^2}$ *where* $X_i$ *is drawn from Lap*$(\frac{2}{\epsilon})$.

*Proof.* If $Y \sim \text{Lap}(b)$, then we know that $\Pr[|Y| \ge t \cdot b] \le e^{-t}$. Therefore,

$$\Pr\left[\max_i |X_i| \ge \frac{10 \log n}{\epsilon}\right] \le \sum_{i=1}^{k} \Pr\left[|X_i| \ge \frac{10 \log n}{\epsilon}\right] \le k \cdot 1/n^5 \le 1/n^2$$

where the last inequality follows from the fact that $k \le n$. $\qquad\qquad\square$

For rest of the proof, we will assume that $|X_i| \le \frac{10 \log n}{\epsilon}$ for all $i$. If any $|X_i| > \frac{10 \log n}{\epsilon}$, then we will not concern ourselves in bounding the privacy loss as it will be absorbed by the $\delta$ parameter. Rounding of $X_i$ to a specific integer value can be seen as a post-processing step, which DP output is immune to. Hence, going forward, we will ignore this minor point.

We prove that our algorithm satisfies Definition 3.1 in two steps: In the first step we assume that adversary does not see the access pattern, and show that output of the algorithm is $(\epsilon, \frac{1}{n^2})$-differentially private.

**Lemma 4.2.** *Our algorithm is* $(\epsilon, \frac{1}{n^2})$*-differentially private with respect to the histogram output.*

*Proof.* We sketch the proof for completeness; see [21] for full proof of the lemma. If the sensitivity of a query is $\Delta$, then we know that the Laplace mechanism, which adds a noise drawn from the distribution $\text{Lap}(\frac{\Delta}{\epsilon})$, is $(\epsilon, 0)$ differentially private. Since the histogram query has sensitivity of 2, $\text{Lap}(\frac{2}{\epsilon})$ is $(\epsilon, 0)$. However, our algorithm outputs the actual histogram without any noise if $\max_i |X_i| > \frac{10 \log n}{\epsilon}$, which we argued in Lemma 4.1 happens with probability at most $\frac{1}{n^2}$. Therefore, our algorithm is $(\epsilon, \frac{1}{n^2})$-differentially private. $\qquad\square$

In the second step we show that memory access patterns of our algorithm satisfies oblivious differential privacy. That is, given any memory access pattern $s$ produced by our algorithm and two neighboring databases $D_1$ and $D_2$ we need to show

$$\Pr[\mathcal{A}(D_1) \to s] \le e^\epsilon \cdot \Pr[\mathcal{A}(D_2) \to s] + \delta$$

To prove this claim we need to set up some notation. Recall that $\hat{D}$ denotes the augmented database with dummy records and fake records. We define *layout* as the actual mapping of the database $\hat{D}$

to the array $a[1], a[2], \ldots, a[T]$. We denote the set of all layouts by $\mathcal{L}$; clearly, $|\mathcal{L}| = T!$, where we assume that dummy/fake records also have unique identifiers. Associated with every layout $\ell$ is a *configuration* $c(\ell)$. A configuration $c(\ell)$ is a $T$-dimensional vector from the set $\{1, 2, \ldots, k+1\}^T$. The $j$th coordinate of $c(\ell)$, denoted by $c_j(\ell)$, simply denotes a type $i \in [k+1]$. Recall that dummy records are of type $k+1$. We extend the histogram function to the augmented database in a natural fashion: $h(\hat{D}) := (n_1, n_2, \ldots, n_{k+1})$, where $n_i$ denotes the number of records of type $i$ in $h(\hat{D})$. The shuffle operation guarantees that mapping of the database $\hat{D}$ to the array $a[1, 2, \ldots, T]$ is uniformly distributed; that is, it picks a layout $\ell \in \mathcal{L}$ uniformly at random.

Given a layout $\ell \in \mathcal{L}$, the memory access pattern produced by our algorithm is completely deterministic. Furthermore, observe that any two layouts $\ell_1, \ell_2 \in \mathcal{L}$, which have the same configuration $c$, lead to same access pattern. Both these statements follow directly from the definition of our algorithm, and the fact that public accesses to the array $b[1, 2, \ldots, k]$ depend only on the type of the records.

Thus, we get the following simple observation.

**Proposition 4.1.** *The mapping $q : \mathcal{C} \to \mathcal{S}$ is a one-to-one mapping between the set of all configurations $\mathcal{C}$ and the set of all access patterns produced by our algorithm $\mathcal{S}$.*

Going forward, we will concern ourselves only with the distribution produced by our algorithm on $\mathcal{C}$ rather than the set $\mathcal{S}$. We will argue that for any two neighboring databases, the probability mass our algorithm puts on a given configuration $c \in \mathcal{C}$ satisfies the definition of DP. The configuration produced by our algorithm depends only on two factors: a) The histogram $h(\hat{D})$ of the augmented database $\hat{D}$ b) Output of the shuffle operation. Up to permutations, a configuration $c$ output by our algorithm is completely determined by the histogram $h(\hat{D})$ produced by our algorithm, which is a random variable. Let $g : \mathbb{R}^{k+1} \to \{1, 2, \ldots, k+1\}^T$ denote the mapping from all possible histograms on augmented databases to all possible configurations of length $T$. However, given a layout of the database $\hat{D}$, the shuffle operation produces a random permutation of the records of the database. This immediately implies the following lemma.

**Lemma 4.3.** *Fix a configuration $c \in \{1, 2, \ldots, k+1\}^T$. Then,*

$$\frac{\Pr\left[\mathcal{A}(D_1) \in c\right]}{\Pr\left[\mathcal{A}(D_2) \in c\right]} = \frac{\Pr\left[\mathcal{A}(D_1) \in g^{-1}(c)\right]}{\Pr\left[\mathcal{A}(D_2) \in g^{-1}(c)\right]}$$

The lemma implies that to show that access patterns produced by our algorithm is $(\epsilon, \delta)$-ODP it is enough to show that distribution on the set of all histograms $\mathbb{R}^{k+1}$ satisfies $(\epsilon, \delta)$-DP. However, the number of dummy elements $n_{k+1}$ in any histogram $h \in \mathbb{R}^{k+1}$ output by our algorithm is completely determined by the random variables $X_1, X_2, \ldots, X_k$. Hence it is enough to show that distribution on the set of all histograms on the first $k$ types satisfies $(\epsilon, \delta)$-DP, which we already argued in Lemma is $(\epsilon, \delta)$-DP.

Now we have all the components to prove the main Theorem 4.4.

*Proof of Theorem 4.4.* For any histogram $h \in \mathbb{R}^{k+1}$, let $\mathsf{truncated}(h)$ denote the histogram restricted to the first $k$ elements. Consider any neighboring databases $D$ and $D'$ and fix a memory access pattern $s$ produced by our algorithm. From Lemma 4.3 and Proposition 4.1 we have that

$$\frac{\Pr\left[\mathcal{A}(D) \to s\right]}{\Pr\left[\mathcal{A}(D') \to s\right]} = \frac{\Pr\left[\mathcal{A}(D) \to g^{-1}(q^{-1}(s))\right]}{\Pr\left[\mathcal{A}(D') \to g^{-1}(q^{-1}(s))\right]}.$$

Since the number of dummy records is completely determined by the random variables $X_1, X_2, \ldots, X_k$, we have

$$\frac{\Pr\left[\mathcal{A}(D) \to g^{-1}(q^{-1}(s))\right]}{\Pr\left[\mathcal{A}(D') \to g^{-1}(q^{-1}(s))\right]} = \frac{\Pr\left[\mathcal{A}(D) \to \mathsf{truncated}(g^{-1}(q^{-1}(s)))\right]}{\Pr\left[\mathcal{A}(D') \to \mathsf{truncated}(g^{-1}(q^{-1}(s)))\right]}$$

which is $(\epsilon, 1/n^2)$-DP from Lemma 4.2. Note that our overall mechanism is $(\epsilon, 1/n^2)$-ODP, since the memory access patterns can be used to construct the histogram output produced by our algorithm. Therefore, we do not lose the privacy budget of $2\epsilon$.

Let us focus on showing that $\max_i |\hat{n}_i - n_i| \leq \log(\frac{k}{\theta}) \cdot \frac{2}{\epsilon}$ with probability at least $1 - \theta$. Consider,

$$\Pr\left[\max_i |\hat{n}_i - n_i| \geq \log(k/\theta) \cdot \frac{2}{\epsilon}\right] \leq \sum_{i=1}^{k} \Pr\left[|\hat{n}_i - n_i| \geq \log(k/\theta) \cdot \frac{2}{\epsilon}\right] \leq k \cdot \frac{\theta}{k} \leq \theta. \quad (1)$$

Now it only remains to bound the running time of the algorithm. First observe that the size of the augmented database is precisely $T$. The shuffle operation takes $O(T \log T / \log \log T)$ and the histogram construction takes precisely $T + k$ time. Therefore the total running time is $O(\tilde{n} \log \tilde{n} / \log \log \tilde{n})$ where $\tilde{n} = \max(n, k \log n / \epsilon)$. $\qquad\square$

## 4.3 Heavy Hitters

As a final example, we consider the problem of finding frequent items, also called the heavy hitters problem, while satisfying the ODP definition. In this problem, we are given a database $D$ of $n$ users, where each user holds an item from the set $\{1, 2, ..., m\}$. In typical applications such as finding the most frequent websites or finding the most frequently used words in a text corpus, $m$ is usually much larger than $n$. Hence reporting the entire histogram on $m$ items is not possible. In such applications, one is interested in the list of $k$ most frequent items, where we define an item as frequent if it occurs more than $n/k$ times in the database. In typical applications, $k$ is assumed to be a constant or sublinear in $n$. The problem of finding the heavy hitters is one of the most widely studied problems in the LDP setting [7, 5]. In this section, we show that the heavy hitters problem becomes very simple in our model. Let $\hat{n}_i$ denote the number of occurrences of the item $i$ output by our algorithm and $n_i$ denotes the true count.

**Theorem 4.5.** *Let $\tau > 1$ be some constant, and let $\epsilon > 0$ be the privacy parameter. Suppose $n/k > \frac{\tau}{\epsilon} \log m$. Then, there exists a $(\epsilon, \frac{1}{m^{\tau-1}})$-ODP algorithm for the problem of finding the top $k$ most frequent items that runs in time $O(n \log n)$. Furthermore, for every item $i$ output by our algorithm it holds that i) with probability at least $(1 - \theta)$, $|\hat{n}_i - n_i| \leq \log(m/\theta) \cdot \frac{2}{\epsilon}$ and ii) $n_i \geq n/k - \log(m/\theta) \cdot \frac{2}{\epsilon}$.*

We remark that the $k$ items output by an ODP-algorithm do not exactly correspond to the top $k$ items due to the additive error introduced by the algorithm. We can use the above theorem to return a list of *approximate heavy hitters*, which satisfies the following guarantees: 1) Every item with frequency higher than $n/k$ is in the list. 2) No item with frequency less than $n/k - 2\log(m/\theta) \cdot \frac{2}{\epsilon}$ is in the list.

We contrast the bound of this theorem with the bound one can obtain in the LDP setting. An *optimal* LDP algorithm can only achieve a guarantee of $|\hat{n}_i - n_i| \leq \sqrt{n \cdot \log(n/\theta) \cdot \frac{\log m}{\epsilon}}$. We refer the reader to [7, 5] for more details about the heavy hitters problem in the LDP setting. For many applications such as text mining, finding most frequently visited websites within a sub-population, this difference in the error turns out to be significant. See experiments and details in [10].

Our algorithm for Theorem 4.5 proceeds as follows: It sorts the elements in the database by type using oblivious sorting. It then initialises an encrypted list $b$ and fills it in while scanning the sorted database as follows. It reads the first element and saves in private memory its type, say $i$, and creates a counter set to 1. It then appends to $b$ a tuple: type $i$ and the counter value. When reading the second database element, it compares its type, say $i'$, to $i$. If $i = i'$, it increments the counter. If $i \neq i'$, it resets the counter to 1 and overwrites the type saved in private memory to $i'$. In both cases, it then appends to $b$ another tuple: the type and the counter from private memory. It proceeds in this manner for the rest of the database. Once finished, it makes a backward scan over $b$. For every new type it encounters, it adds $\mathrm{Lap}(2/\epsilon)$ to the corresponding counter and, additionally, extends the tuple with a flag set to 0. For all other tuples, a flag set to 1 is added instead. It then sorts $b$: by the flag in ascending order and by differentially-private counter values in descending order.

Let $n^*$ be the number of distinct elements in the database. Then the first $n^*$ tuples of $b$ hold all the distinct types of the database together with their differentially-private frequencies. Since these elements are sorted, one can make a pass, returning the types of the top $k$ most frequent items with the highest count (which includes the Laplace noise). Although this algorithm is not $(\epsilon, 0)$-ODP, it is easy to show that it is $(\epsilon, \frac{1}{m^{\tau-1}})$-ODP when $n/k > \tau \log m$, which is the case in all applications of the heavy hitters. Indeed, in most typical applications $k$ is a constant. We are now ready to prove the above theorem.

*Proof of Theorem 4.5.* The running time of the algorithm is dominated by oblivious sorting and is $O(n \log n)$ if using AKS sorting network. The algorithm accesses the database and list $b$ independent of the data, hence, the accesses are also obliviously differentially private. In the rest of the proof, we show that the output of the algorithm is differential-private.

Fix any two neighboring databases $D$ and $D'$. Suppose distinct$(D)$, distinct$(D')$ denote the number of distinct items that appear in databases $D$ and $D'$. If distinct$(D) =$ distinct$(D')$, then the distinct items appearing in the two databases should be the same as $D$ and $D'$ differ only in 1 row. In this case, the histogram constructed on the distinct items by our algorithm is differentially private since it is an oblivious implementation of the simple private histogram algorithm [21]. Any post-processing done on top of a differentially private output, such as reporting only top $k$ items does not violate the DP guarantee. Hence our overall algorithm is DP.

Now consider the case when distinct$(D) \neq$ distinct$(D')$. In this case, let $i^*$ denote the item that appears in $D'$ but not in $D$. Clearly, $n_{i^*} = 1$ in $D'$ and $n_{i^*} = 0$ in $D$. Let $I$ denote the set of items that are common in the datasets $D$ and $D'$. If one restricts the histogram output to the set $I$, it satisfies the guarantees of DP. However, our algorithm never reports $i^*$ in the list of heavy hitters on the database $D$. On the other hand, there is a non-zero probability that our algorithm reports $i^*$ in the list of top $k$ items. This happens if the Laplace noise added to $i^*$ is greater than $n/k \geq \frac{\tau}{\epsilon} \log m$, which occurs with the probability at most $\frac{1}{m^\tau}$. Since there are at most $m$ items, by the union bound the probability of this event happening is $\frac{1}{m^{\tau-1}}$. $\square$

**Frequency Oracle Based on Count-Min Sketch**  Another commonly studied problem in the context of heavy hitters is the frequency oracle problem. Here, the goal is to answer the number of occurrences of an item $i$ in the database. While this problem can be solved by computing the answer upon receiving a query and adding Laplace noise, there is a simpler approach which might be sufficient in many applications. One can maintain a count-min sketch, a commonly used algorithm in the streaming literature [15], of the frequencies of items by making a single pass over the data. An interesting aspect of this approach is that entire sketch can be maintained in the private memory, hence one does not need to worry about obliviousness. Further, entire count-min sketch can be released to the data collector by adding Laplace noise. An advantage of this approach is that the data collector can get the frequencies of any item he wants by simply referring to the sketch, instead of consulting the DP algorithm. It would be interesting to find more applications of the streaming techniques in the context of ODP algorithms.

## Footnotes

[2]"Introducing Azure confidential computing", accessed October 26, 2019.

[3]"Introducing Asylo: an open-source framework for confidential computing", accessed October 26, 2019.

[4]This should not be confused with vulnerabilities introduced by floating-point implementations [37].

[5]Here, data collector runs algorithms on premise. See appendix for restrictions in the cloud setting.

[6]Data stored in external memory is highlighted in grey. Recall that it is encrypted.

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

# A Cloud Computing Setting

If the framework is hosted on the cloud, we also consider a second adversary $A_C$ who is hosting the framework (e.g., malicious cloud administrator or co-tenants). Since $A_C$ has access to the infrastructure of the framework its adversarial capabilities are the same as those of the adversarial data collector $A_D$ in §1. However, in this case, in addition to protecting user data from $A_D$, the result of the computation also needs to be protected. We note that if $A_D$ and $A_C$ are colluding, then the cloud scenario is equivalent to the on-premise one. If they are not colluding, $A_C$ may still be able to gain access to the framework where its outsourced computation is performed. It can do so either by using malware or as a malicious co-tenant [48].

In this setting, our framework remains as described in §3 with the exception that algorithms that run inside of the TEE need to be data-oblivious per Definition 2.2 in order to hide the output from $A_C$. Since Definition 2.2 is stronger than Definition 3.1 when output is not revealed, user data is also protected from $A_D$.

# B  ORAM-based Histogram Algorithm

In this section we outline a standard differentially private algorithm for histogram computation and its ORAM-based transformation to achieve ODP. We note that the algorithm in §4.2 is more efficient than this transformation for large values of $k$.

Algorithm 3 is the standard differentially private algorithm from [21]. It is not data-oblivious since accesses to the histogram depend on the content of the database and reveal which records have the same type. (See Figure 1 (right) for a visualization.)

Algorithm 4 is a data-oblivious version of Algorithm 3. It uses oblivious RAM (see §2.2) to hide accesses to the histogram. In particular, we use $\mathsf{ORAM}(b)$ to denote the algorithm that returns a data-oblivious data structure $\tilde{b}$ initialized with an array $b$. $\tilde{b}$ supports queries $\tilde{b}.\mathsf{read}(i)$ and $\tilde{b}.\mathsf{write}(i, \mathsf{data}_i)$ for $i \in [1, k]$. That is, it returns $b.\mathsf{read}(i)$ (or similarly stores $\mathsf{data}_i$ in the $i$th index of $b$ with write) while hiding $i$. The performance of the resulting algorithm depends on the underlying oblivious RAM construction. For example, if we use the scheme by Asharov *et al.* [3], it makes $O(n \log k)$ accesses to external memory, or $O(n(\log k)^2)$ if we use Path ORAM [54] that has much smaller constants.

The histogram problem can be solved also using two oblivious sorts, similar to the heavy hitters algorithm described in §4.3 but returning all $n^*$ types and their counters since $k = n^*$ when histogram support is known. The overhead of this method is the overhead of the underlying sorting algorithm.

| **Algorithm 3** DP histogram [21] $\mathcal{M}_{\mathsf{hist}}^{\mathsf{DP}}(D, k)$ | **Algorithm 4** ORAM-based DP histogram $\mathcal{M}_{\mathsf{hist}}^{\mathsf{ODP}}(D, k)$ |
|---|---|
| $b \leftarrow \{0\}^k$ <br><br> **for** $r \in D$ **do** <br>    $i \leftarrow \mathsf{get\_type}(r)$ <br>    $b_i \leftarrow b_i + 1$ <br><br> **end for** <br> **for** $i \in 1 \dots k$ **do** <br>    $\hat{b}_i \leftarrow b_i + \mathrm{Lap}(\frac{2}{\epsilon})$ <br> **end for** <br> return $\hat{b}$ | $b \leftarrow \{0\}^k$ <br> $\tilde{b} \leftarrow \mathsf{ORAM}(c)$ <br> **for** $r \in D$ **do** <br>    $i \leftarrow \mathsf{get\_type}(r)$ <br>    $b_i \leftarrow \tilde{b}.\mathsf{read}(i)$ <br>    $\tilde{b}.\mathsf{write}(i, b_i + 1)$ <br> **end for** <br> **for** $i \in 1 \dots k$ **do** <br>    $\hat{b}_i \leftarrow \tilde{b}.\mathsf{read}(i) + \mathrm{Lap}(\frac{2}{\epsilon})$ <br> **end for** <br> return $\hat{b}$ |

# C Comparison of Data-Oblivious Algorithms

Table 1: Asymptotical performance of several data-oblivious algorithms on arrays of $n$ records, $c \geq 2$ is a constant, $m$ is the size of private memory, $k$ is the number of types in a histogram, and $\tilde{n} = \max(n, k \log n / \epsilon)$.

| | **Algorithm** | **Private Memory** ($m$) | **Overhead** |
|---|---|---|---|
| RAM | Path ORAM [54] | $\omega(\log n)$ | $(\log n)^2$ |
| | Kushilevitz *et al.* [34] | 1 | $\frac{(\log n)^2}{\log \log n}$ |
| | Asharov *et al.* [3] | 1 | $\log n$ |
| Sort | AKS Sort [1] | 1 | $n \log n$ |
| | Batcher's Sort [8] | 1 | $n(\log n)^2$ |
| Shuffle | Melbourne Shuffle [40, 44] | $\sqrt[c]{n}$ | $cn$ |
| | | $\omega(\log n)$ | $n \frac{\log n}{\log m}$ |
| Histogram | In private memory, $k = O(m)$ | $k$ | $n$ |
| | ORAM-based (Algorithm 4) | 1 | $n \log k$ |
| | Oblivious Sort-based (§B) | 1 | $n \log n$ |
| | ODP Histogram (§4.2, Algorithm 1) | $\omega(\log n)$ | $\tilde{n} \frac{\log \tilde{n}}{\log \log \tilde{n}}$ |