[Reviews · NeurIPS 2019]

Reviewer 1



Content: The authors’ introduce a framework for practical DP computation using a secure environment with limited memory and a notion they call “oblivious differential privacy”. They give several algorithms for computation in this framework and show that they can achieve accuracy guarantees close to CDP, but have a trust structure that is more similar to LDP. This is an important area of research as companies grow increasingly wary of holding sensitive user data in the clear. This paper makes some interesting conceptual contributions to this discussion. The statistical tasks they present algorithms for are very common and useful tasks. These tasks are also examples of problems where there is a significant gap in performance between LDP and CDP algorithms. Originality: As far as I am aware, ODP is a new concept, although the authors’ mention that it has appeared in concurrent work. The use of oblivious shuffling to hide dummy and fake records in the histogram algorithm is interesting and the proof is non-trivial. The most interesting aspect of the definition of ODP to me is that it is actually a weaker condition than the algorithm being data oblivious. However, the authors’ don’t really seem to take advantage of this and most of the algorithms rely on heavy machinery from the oblivious computations and streaming algorithms literature. The histogram and heavy hitters algorithms use oblivious shuffling, which I understand (in my limited understanding of oblivious algorithms) to have more overhead than non-oblivious shuffling. Can you get simpler algorithms by leveraging the fact that we don’t require total obliviousness? The other idea that I haven’t seen much in the literature is performing DP mechanisms with limited memory. This is an interesting direction although again in this paper, it seems to be reduced to using existing streaming algorithms. Quality: The computations and proofs of the main theorems appear correct. All the definitions are sensible and present interesting ideas. Any assumptions made seems reasonable. I would like further clarification on how much overhead is required to set up the secure environment. Also, the encryption/decryption step appears to be ignored when discussing the run-time of the oblivious algorithms in the secure environment? How does the overhead in this setting compare to that of the “analyses, shuffle and analyse” framework? Clarity: The paper is well-written. The motivation and main results are clear. The diagram in Figure 1 (left) seems incomplete. From my understanding of the set-up, the algorithm can also access encrypted data stored outside the trusted execution environment, provided it is accessed in an oblivious way? This is an important distinction since data stored inside the secure environment does not need to be accessed in an oblivious manner, hence the use of streaming algorithms. Significance: This work explores bridging the gap between the local DP and central DP. It attempts to simulate the trust model of LDP using a secure computation environment, but also provide accuracy guarantees that are similar to CDP. This is an important area of research that has started to receive significant attention in the literature. The trust model of LDP can be more appealing not only to users, by also to data aggregators as it protects them from unintentional data leakages. However, for many tasks there is a significant gap between accuracy bounds for LDP and CDP algorithms. From a legal perspective, TEEs may protect companies from being liable for protecting users data. I think this work explores two exciting directions for DP research: 1. The idea of weakening definitions like obliviousness (and, as has occurred elsewhere in the literature, secure multi-party computation) to allow DP leakage is an interesting idea that has a lot of potential. I can see the definition of ODP in this paper propelling a lot of interesting research. 2. The idea of performing DP computations in a secure environment with limited memory. Unlike in the typical streaming setting, this allows for the potential of storing DP statistics outside of the secure environment to be used later. I am curious whether, and to what degree, this additional freedom allows one to improve the accuracy of streaming algorithms for DP. I can see this idea sparking interest in members of the community. Minor comments: - When surveying previous works, the authors’ mention recent work on the “analyses, shuffle and analyse” framework. They mention that this work can be seen as only shifting the trust boundary since the user has to trust that the anonymisation primitives are effective. The authors’ allude several times to how the secure computation environment has more safeguards, e.g. they discuss that the users can verify that they are communicating with a specific piece of software. However, I would be interested to hear more about how trusting an anonymisation primitive is different to trusting a secure environment? I don’t see this as a failing of either technique, but would be interested in hearing more about the differences between the trust structures.

Reviewer 2



There are two main models for differential privacy: the central model which assumes a trusted aggregator which can see and compute on the data in the clear but adds noise to the output in order to provide differential privacy, and the local model where each party adds noise locally to its inputs and the aggregator no longer needs to be trusted. While the local model provides much stronger privacy guarantees, unfortunately it achieves much lower utility which is not sufficient in many practical settings. This paper presents a new model which assumes trusted processors that essentially execute the central DP mechanism but do not have sufficient memory to fit all data. In order to avoid the leakage from memory accesses that have to happen outside the trusted environment, the authors propose to use oblivious algorithms that have access patterns independent of their inputs. This conceptual composition of TTP and memory oblivious algorithms has been considered in many previous works that compose trusted execution with limited memory with oblivious memory techniques such as ORAM or other oblivious algorithms. Thus the contribution of this work has to be searched in the particular oblivious instantiations of DP algorithms. The paper presents three oblivious DP algorithms in the TTP framework: algorithm for counting distinct items in a database, computing a histogram and computing heavy hitters. I found the proposed constructions quite direct following previous approaches. For the first problem of counting the number of distinct items, the authors use a streaming approach that relies on a sketching technique to maintain an approximate count in private memory and the authors propose to add Laplacian noise to these counts. The solution for histogram computation uses the observation that comes from the work of Mazloom and Gordon to add fake and dummy records to the data, obliviously shuffle and then just do the counts with a linear scan. In the description of this protocol I was confused by the fact that at one place the authors claimed that oblivious shuffle is expensive and then use oblivious shuffle in step 4 of their protocol. The third protocol for heavy hitters is not even described in the main body of the paper. This construction is also direct and at the same time costly with two oblivious sorting invocations: it first sorts all elements, then computes counts with a linear scan, then it sorts the counts and adds noise to them, and then finally extracts the top counts for the heavy hitters. The constructions of this paper seem to follow mostly from previous work. I did not also see a convincing analysis for the efficiency of the proposed constructions in a TTP architecture especially given the oblivious sorts and shuffles used in the constructions. Related work: there is a new paper that extends the results from the shuffle DP model to larger values of epsilon: https://arxiv.org/abs/1903.02837. There are ORAM constructions with better amortized overhead than the ones claimed in this paper: https://eprint.iacr.org/2018/373.pdf, https://eprint.iacr.org/2018/892.

Reviewer 3



Summary: The paper presents a privacy system that works with trusted enclaves (e.g. SGX) to collect data and run (globally) differentially private algorithms. Practical deployments of differential privacy usually are in the local model, where data is privatized prior to aggregation on the server. The benefit of the local model is that users can ensure their data is privatized before submission, thus no trust is implied. However, the local model of DP suffers from a significant utility loss, even for moderate privacy levels. This paper provides a solution to use a combination of Trusted Execution Environments and global differential privacy to benefit from the better utility of the global model while not requiring implicit user trust in the system. Critique: This paper addresses a very important problem faced by privacy practitioners that want to obtain the accuracy of differential privacy in the global model while also not needing to store the users’ data in an unencrypted format or requiring users’ trust. This work opens the door for lots of future work that combines security hardware and privacy software, which is a natural combination but not really explored before. The future work will be both theoretical, developing new ODP algorithms, as well as application driven, showing results for particular use cases. My main critique is that the paper quickly addresses other side channel attacks, beyond memory access patterns, to say that the framework can be extended. However, how do the algorithms change if we were to include both timing attacks and memory access patterns? Are there impossibility results with trying to protect against several side channel attacks while ensuring data-obliviousness? Minor Comments: some missing “the”s in the paper. To me, the problem of heavy hitters is more interesting that histograms, so I would like to see more detail about the heavy hitters algorithm in the camera ready instead of the histogram use.

[Author Response · NeurIPS 2019]

We thank the reviewers for their positive feedback, and address their main concerns below. Given the opportunity, we
will address other concerns in the final version. We are grateful to Reviewers #1 and #3 for seeing the potential of our
work to spark future research in the intersection of differential privacy, TEEs and oblivious algorithms.

**Reviewer #1:** *Q: What is the overhead for setting up the secure environment, the encryption/decryption step. How*
*does it compare to that of the LDP+shuffle [55,56] and ESA of [8]* **A:** Setting up an enclave is a one time cost and is
proportional to the size of the code and data, giving a linear overhead. The overhead of encryption/decryption is also
linear. It is hard to compare overhead of our framework to that of [55,56] as implementation of shuffle is largely left
unexplained in [55, 56]. One natural way to implement shuffle step in [55] is indeed to use TEEs, then the overhead
of these frameworks should be comparable. If implemented using mixnets [56], the overhead might even be higher.
However, this is a very good point, and we will add this in the final version.

*Q: How trusting an anonymization primitive is different to trusting a secure environment?* **A:** Anonymization using
mixnets [56] will require assumption on non-collusion between the servers. If anonymization is implemented via
TEEs, then the trust model would be the same as ours. While LDP+shuffle idea in [55,56] is mathematically elegant,
DP algorithms inside TEEs come with two major technical advantages: 1) One can use DP algorithms in the central
model. Consider for example using private multiplicative weights algorithm for answering exponentially many linear or
counting queries. This has no parallels in the LDP+shuffle model. Even with amplification result one can only answer
polynomially many queries if we want to achieve same level of accuracy as the central model. 2) Given the growing
software support for TEEs by Google, IBM, Microsoft, DP+TEEs approach, arguably, seems closer to adoption in
practice.

*Q: Do the authors' conjecture that the weakening of the requirements of the algorithms will result in faster/more*
*accurate algorithms?* **A:** We agree with your intuition. We also believe ODP definition should allow us to design faster
algorithms for DP problems than simply combining with the stronger notion of oblivious algorithms. We plan to explore
this direction in the future. The accuracies achieved by our algorithms for all the 3 problems considered match the
trusted curator model asymptotically.

**Reviewer #2: Q:** *Composition of TTP and memory oblivious algorithms has been considered in many previous works*
**A:** As we mention in our paper, we agree that composition of TEEs (or TTP) with oblivious algorithms, in general, is
not new. If this was not clear, we will make sure that this is stated more clearly in the final version. First, no paper
earlier to our work has suggested *running central DP* algorithms within a TEE (*). This idea leads to combining
oblivious algorithms and memory-restricted algorithms for the design of differentially private algorithms, and is a new
contribution. Even considering all the concurrent/other works, our ODP definition is new: we consider an adversary that
*gains access to the output of the computation*. We then ask the question of how to securely and efficiently instantiate
the global DP setting using TEEs. We observe that since DP output is revealed, the access patterns do not have to be
strictly oblivious, motivating our new definition. The opposite direction, applying DP to oblivious algorithms, has been
explored in independent work [11]. [8] explores using oblivious shuffle for anonymization, and is technically different
from our ODP definition. We cannot prove the independence of our work to (e.g., [8,11]) or (*) to preserve anonymity.

*Q: In Histogram protocol, at one place the authors claimed that oblivious shuffle is expensive and then use oblivious*
*shuffle in step 4.* **A:** We mention that *oblivious sort* is expensive so we use the *shuffle* in step 4.

*Q:The constructions of this paper seem to follow mostly from previous work.* **A:** While some of the individual
components of our algorithms have appeared before (and we cite), combining ideas from oblivious algorithms literature
to DP, and design of DP algorithms with limited memory are both new, and have not appeared before (as also noted
by Reviewer #1). Further, some of the previous works is parallel. *Q: I did not also see a convincing analysis for*
*the efficiency of the proposed constructions in a TTP architecture.* **A:** We have provided full proofs of theorems
in the supplementary material. Due to page limit, we could not give full proofs in the main body or discuss all the
improvements our theorems imply. For example, from Theorem 4.4 our histogram construction is more efficient than
oblivious constructions for larger values of $k$ (see also Table 1 in the Appendix).

**Q:** *Better analysis of the costs of the protocols and the gains compared with a trusted aggregator model, or the shuffle*
*and compute model.* **A:** Our algorithms achieve same level of accuracy guarantees as that of trusted aggregator model,
as can be seen from our theorems. The cost of our framework lies in the increased running time not accuracy. Compared
to LDP+shuffle model, from theorems in [55,56], our accuracy guarantees are better.

**Q:** *Related Work suggestions* **A:** Thank you for related work suggestions. We will cite these papers appropriately and
update Table 1. From a quick reading, it appears that privacy blanket paper still does not help us achieve accuracy
guarantees of the trusted aggregator model, as it will require larger values of epsilon.

**Reviewer #3: Q:** *Address other side channel attacks a little more thoroughly and explain how to design algorithms.*
**A:** Our Definition 3.1 extends to other side channels, and we will make it more clear in the final version. However,
design of DP algorithms incorporating *all* side channels is challenging and is an interesting research direction on its
own (even when DP is not considered). We will discuss our ideas for preventing timing attacks in the camera-ready
version if given an opportunity. We will also add more details about the heavy hitters in the final version.

[Meta-Review · NeurIPS 2019]

This paper proposes “oblivious differential privacy” (ODP), a notion of privacy-preserving computation taking into account the leakage that memory access patterns can reveal about the input data when a differentially private mechanism is ran inside a trusted execution environment (TEE). The paper presents algorithms for several basic tasks in this model; most of these algorithms use ideas and building blocks developed elsewhere. The main value of this paper is proposing a robust definition privacy capable of taking into account particularities of the computing architecture where the mechanism is executed, and showing that several basic computations can be performed efficiently and privately in this model. ODP has the potential to become a standard model for research in privacy-preserving computation inside TEEs.